# An Overview of Systematic Reviews of Polymerase Chain Reaction (PCR) for the Diagnosis of Invasive Aspergillosis in Immunocompromised People: A Report of the Fungal PCR Initiative (FPCRI)—An ISHAM Working Group

**DOI:** 10.3390/jof9100967

**Published:** 2023-09-26

**Authors:** Mario Cruciani, P. Lewis White, Rosemary A. Barnes, Juergen Loeffler, J. Peter Donnelly, Thomas R. Rogers, Werner J. Heinz, Adilia Warris, Charles Oliver Morton, Martina Lengerova, Lena Klingspor, Boualem Sendid, Deborah E. A. Lockhart

**Affiliations:** 1Fungal PCR Initiative (FPCRI), 37100 Verona, Italy; 2Public Health Wales, Microbiology Cardiff, UK and Centre for Trials Research, Division of Infection and Immunity, Cardiff University, Cardiff CF14 4XW, UK; lewis.white@wales.nhs.uk; 3School of Medicine, Cardiff University, Cardiff CF10 3AT, UK; barnesra@cardiff.ac.uk; 4Department of Internal Medicine II, University Hospital of Würzburg, 97070 Würzburg, Germany; 5European Aspergillus PCR Initiative, Nijmegen, The Netherlands; 6Discipline of Clinical Microbiology, Trinity College Dublin, St. James’s Hospital Campus, LS9 7TF Dublin, Ireland; rogerstr@tcd.ie; 7Medicine Clinic II, Caritas Hospital Bad Mergentheim, 97980 Bad Mergentheim, Germany; 8MRC Centre for Medical Mycology, University of Exeter, Exeter EX4 4QJ, UK; a.warris@exeter.ac.uk; 9School of Science, Western Sydney University, Campbelltown Campus, Campbelltown, NSW 2751, Australia; o.morton@westernsydney.edu.au; 10Central European Institute of Technology, Masaryk University, 60177 Brno, Czech Republic; 11Department of Laboratory Medicine, Karolinska Institutet, 17177 Stockholm, Sweden; lena.klingspor@ki.se; 12Inserm U1285, CNRS UMR 8576, UGSF, CHU Lille, Laboratoire de Parasitologie-Mycologie, University of Lille, 59000 Lille, France; boualem.sendid@univ-lille.fr; 13Institute of Medical Sciences, School of Medicine Medical Sciences and Nutrition, University of Aberdeen, Aberdeen AB24 3FX, UK

**Keywords:** systematic review, meta-analysis, umbrella review, *Aspergillus fumigatus*, invasive aspergillosis, diagnosis, polymerase chain reaction (PCR)

## Abstract

This overview of reviews (i.e., an umbrella review) is designed to reappraise the validity of systematic reviews (SRs) and meta-analyses related to the performance of *Aspergillus* PCR tests for the diagnosis of invasive aspergillosis in immunocompromised patients. The methodological quality of the SRs was assessed using the AMSTAR-2 checklist; the quality of the evidence (QOE) within each SR was appraised following the GRADE approach. Eight out of 12 SRs were evaluated for qualitative and quantitative assessment. Five SRs evaluated *Aspergillus* PCR on bronchoalveolar lavage fluid (BAL) and three on blood specimens. The eight SRs included 167 overlapping reports (59 evaluating PCR in blood specimens, and 108 in BAL), based on 107 individual primary studies (98 trials with a cohort design, and 19 with a case−control design). In BAL specimens, the mean sensitivity and specificity ranged from 0.57 to 0.91, and from 0.92 to 0.97, respectively (QOE: very low to low). In blood specimens (whole blood or serum), the mean sensitivity ranged from 0.57 to 0.84, and the mean specificity from 0.58 to 0.95 (QOE: low to moderate). Across studies, only a low proportion of AMSTAR-2 critical domains were unmet (1.8%), demonstrating a high quality of methodological assessment. Conclusions. Based on the overall methodological assessment of the reviews included, on average we can have high confidence in the quality of results generated by the SRs.

## 1. Introduction

Invasive aspergillosis (IA) is a life-threatening opportunistic invasive mold disease of the immunocompromised host [1] and, as such, requires early diagnosis and prompt systemic antifungal treatment to enhance survival [2]. Consequently, there is an urgent need for new diagnostic tools and optimization of the use of existing tests individually, or in combination, to better complement antifungal treatment [3,4]. *Aspergillus* polymerase chain reaction (PCR) testing of blood and respiratory samples has recently been included in the second revision of the EORTC/MSGERC definitions for classifying invasive fungal disease [5,6].

Due to the large number of published papers on *Aspergillus* PCR (>2500 papers to date (PubMed search for *Aspergillus* PCR)), there is a significant amount of clinical data available, and a number of systematic reviews (SRs) and meta-analyses on the performance of PCR for the diagnosis of IA have been published [7,8]. Their conclusions show extensive heterogeneity among studies in terms of design, conduct, and reporting. The current study is an overview of SRs, termed an umbrella review, aimed at reappraising the validity of the conclusions of SRs and the diagnostic accuracy of PCR-based tests on blood and respiratory specimens for the diagnosis of IA in immunocompromised patients published in meta-analyses. The decision to perform this overview was based on the continuing importance of the review question, and on the availability of new data from SRs/meta-analyses. Increasing the number of studies can improve precision of effect estimates, allowing additional comparisons or subgroup analyses to be performed [9]. An umbrella review is a review of reviews, and only considers other systematic reviews as eligible for inclusion (in other words, the unit of analysis for overview of reviews is the systematic review/meta-analysis and not individual patient data). An umbrella review collects evidence from multiple existing reviews and provides perhaps the highest levels of evidence. In this umbrella review, we have also applied new review methods such as the AMSTAR-2 tool, and a GRADE assessment, with the aim of enhancing the existing results in terms of the certainty of the review’s findings [10].

## 2. Material and Methods

The protocol of this overview of reviews is available on the International Prospective Register of Systematic Reviews (PROSPERO) with the registration number CRD42021259625. There were no amendments from the pre-specified criteria reported in the protocol throughout the review process. The results are reported according to Preferred Reporting Items for a Systematic Review and Meta-analysis of Diagnostic Test Accuracy Studies (PRISMA-DTA) [11].

### 2.1. Inclusion and Exclusion Criteria

This overview includes SRs evaluating clinical trials (i) aimed at comparing the performance of PCR tests with reference to the consensus definitions of IA published by the European Organization for Research and Treatment of Cancer/Mycoses Study Group (EORTC/MSGERC [6,12,13]); (ii) reporting data on false-positive, true-positive, false-negative, and true-negative results for the diagnostic tests under investigation separately; and (iii) evaluating the test(s) in cohorts of patients from a relevant clinical population, defined as a group of individuals at high risk for IA.

### 2.2. Search Strategy

Relevant studies in three bibliographic databases (Embase, PubMed, and Cochrane library) were searched up to March 2023. The searches were performed without language restriction using Medical Subjects Heading: (“Aspergillosis/Aspergillus” or “Invasive fungal Infection”) AND “diagnosis” AND (“systematic review” OR “meta-analysis). Furthermore, reference lists of the reviews were checked to identify potentially eligible studies not captured by the electronic literature search.

### 2.3. Study Selection and Data Extraction

All titles were screened by two independent assessors (MC and LW). Eligibility assessment was initially based on the title or abstract, and on the full text, when required. Full texts of potentially eligible articles were obtained and assessed independently by two reviewers (MC and LW) against the stated inclusion criterion. The study selection decision of each reviewer was compared for concordance. The two assessors also independently extracted quantitative and qualitative data from each selected study, with disagreements resolved through discussion and through the opinion of a third reviewer (RB). Findings are presented in tabular format with Appendix A. Tabulation of results includes the following: first author name and year of publication, clinical setting, number and design of studies included in the SR, index test and reference standard, subgroup analyses, and the main conclusion of the review as reported by authors.

### 2.4. Assessment of Methodological Quality of Systematic Reviews

We used the AMSTAR-2 critical appraisal checklist, a tool for SRs that includes randomized or non-randomized studies, or both [14]. The tool is suitable for reviews of intervention, but can also be adapted to explore SRs of diagnostic tests. It includes 16 domains, of which 7 are considered critical, relating to the research question, review design, search strategy, study selection, data extraction, justification for excluded studies, description of included studies, risk of bias, sources of funding, meta-analysis, heterogeneity, publication bias, and conflicts of interest. Two review authors (MC, LW) independently assessed the quality of evidence and the methodological quality of the SRs. We resolved discrepancies through discussion or, if needed, through a third review author (RB). Reviews were not excluded based on AMSTAR 2 ratings, but the ratings were considered in interpretation of the results.

### 2.5. Summary of the Evidence, Subgroups Analisis, and Appraisal of the Quality of Evidence

For the quantitative synthesis, the sensitivity and specificity were reported (when available) with the 95% confidence intervals (CIs), as stated in the individual reviews. Where available, other measures of diagnostic accuracy such as predictive values, likelihood ratios, and diagnostic odds ratio were reported. Moreover, the impacts of several variables on the diagnostic performance of PCR were evaluated, as reported in the SRs by subgroups analysis or meta-regression. To this end, we focused on differences in study design (e.g., cohort vs. case−control studies), patient selection (e.g., hematology vs. other at-risk patients), variations in the index test and reference standard, and the use of antifungal agents.

The quality of evidence was appraised following the GRADE approach (Grades of Recommendation, Assessment, Development, and Evaluation) [15,16]. Whenever available, the grading of the quality of evidence reported in the SRs was considered to define the quality of evidence. When grading of evidence was not reported by the authors of the study, the GRADE approach was applied based on the information available in the individual review. Studies can be downgraded because of concerns over the risk of bias, indirectness (applicability of the results to the question), inconsistency (heterogeneity between study results), imprecision (low number of studies and/or participants), and publication bias [16]. The GRADE approach has four levels of certainty: very low (the true effect is probably markedly different from the estimated effect), low (the true effect might be markedly different from the estimated effect), moderate (the true effect is probably close to the estimated effect), and high (the true effect is similar to the estimated effect) [16].

## 3. Results

The electronic and manual search retrieved 118 references. The Preferred Reporting Items for SRs and Meta-Analyses (PRISMA) flow diagram is reported in Figure 1.

At the first stage of screening titles and abstracts, 14 references were selected [7,8,17,18,19,20,21,22,23,24,25,26,27,28]. After the full texts were examined with regards to eligibility (i.e., inclusion and exclusion criteria), twelve records were considered for this umbrella review [17,18,19,20,21,22,23,24,25,26,27,28] but, to avoid the inclusion of duplicate records, data were extracted from nine records (eight SRs), with the exclusion of three reviews [17,18,19], due to the availability of updated versions which were included [23,26]. Two previous overviews of reviews were also excluded [7,8].

### 3.1. Description of the Studies

Of the eight SRs included in the overview, five evaluated PCR on bronchoalveolar lavage fluid (BAL) specimens, with three on blood specimens. Four reviews focused exclusively on PCR, while four compared the diagnostic performance of PCR test to galactomannan and/or beta-d glucan [22,23,24,25]. The eight SRs included 167 overlapping reports (59 evaluating PCR in blood specimens, and 108 in BAL), based on 107 individual primary studies. The 107 primary studies included 98 trials with a cohort design (47 with blood specimens, 41 with BAL specimens), and 19 with a case−control design (3 with blood specimens, 16 with BAL specimens). The main characteristics of the SRs included are summarized in Table 1.

### 3.2. Methodological Quality of the SRs with the AMSTAR-2

Three of the SRs met all the AMSTAR-2 methodological requirements, and three SRs partially met one or two of the methodologic requirements, while fully meeting the rest (Table 2). Two SRs had one unmet methodologic requirement, plus between two and six partially met requirements. Of the 120 methodological requirements assessed (Domain 10 excluded) across all eight studies, a total of 12 (10%) methodological requirements were only partially met, with only 1.7% unmet and 88.3% (106/120) of domains fully met. Of the seven critical domains (see footnote of Table 2), only one (1.8% of critical domains across all the SRs) item was judged to have been unmet, and six (10.7%) partially met, leaving 87.5% of critical domains fully met across all SRs. Based on the overall methodological assessment and considering an unmet requirement as an indicator of lower confidence, there is high confidence in the results generated from the majority (75%) of SRs included in the overview [14]. For this overview, one item (sources of funding for the studies included in the review) was not included, given that the large majority of primary studies evaluated were in-house PCR tests and the process of SR and meta-analysis was independent of financial support.

Amstar-2 domains:Did the research questions and inclusion criteria for the review include the components of PICO (patients, index test, comparator, accuracy as outcome)?Did the report of the review contain an explicit statement that the review methods were established prior to the conduct of the review and did the report justify any significant deviations from the protocol?Did the review authors explain their selection of the study designs for inclusion in the review?Did the review authors use a comprehensive literature search strategy?Did the review authors perform study selection in duplicate?Did the review authors perform data extraction in duplicate?Did the review authors provide a list of excluded studies and justify the exclusions?Did the review authors describe the included studies in adequate detail?Did the review authors use a satisfactory technique for assessing the risk of bias (RoB) in individual studies that were included in the review?Did the review authors report on the sources of funding for the studies included in the review?If meta-analysis was performed did the review authors use appropriate methods for statistical combination of results?If meta-analysis was performed, did the review authors assess the potential impact of RoB in individual studies on the results of the meta-analysis or other evidence synthesis?Did the review authors account for RoB in individual studies when interpreting/discussing the results of the review?Did the review authors provide a satisfactory explanation for, and discussion of, any heterogeneityIf they performed quantitative synthesis did the review authors carry out an adequate investigation of publication bias (small study bias) and discuss its likely impact on the results of the review?Did the review authors report any potential sources of conflict of interest, including any funding they received for conducting the review?

Although AMSTAR 2 consists of 16 items, critical domains include items 2, 4, 7, 9, 11, 13, and 15. Item 10 was not assessed (na).

### 3.3. Summary of the Performance of PCR for the Diagnosis of Invasive Aspergillosis

Mean sensitivity and specificity and 95% CIs (if available) of PCR as calculated for each meta-analysis are summarized in Table 3. In BAL specimens, the mean sensitivity ranged from 0.57 to 0.91, and the mean specificity from 0.92 to 0.97. The level of certainty of these findings for studies within individual SRs was considered low to very low, mostly due to the risk of bias (selection bias in case−control studies) and inconsistency (due to heterogeneity) in primary studies included in the reviews.

In blood specimens (whole blood or serum), the mean sensitivity ranged from 0.57 to 0.84, and the mean specificity from 0.58 to 0.95. The results of two studies that performed subgroup analysis according to the number of specimens required to define the test positive (a single positive specimen or ≥2 consecutive positive specimens) were consistent [23,26]. In these two SRs, mean sensitivity values were 0.79 and 0.84 for 1 positive test, and 0.57 and 0.59 for ≥2 positive tests, and mean specificity values were 0.79 for 1 positive test, and 0.93 and 0.95 for ≥2 positive tests. Lower specificity (0.58) was found in the review of PCR performance in pediatric patients [25]. We graded the level of certainty of these findings as being low to moderate. The overall quality of the evidence according to the GRADE assessment was very low for three SRs, low for four SRs, and moderate for one SR (Table 3).

### 3.4. Other Measures of Diagnostic Performance and Subgroup Analyses 

In BAL specimens, positive and negative likelihood ratios from three reviews [20,21,28] ranged from 10.4 to 11.9 and 0.10 to 0.27, respectively. In one review, positive and negative predictive values were 81.6 and 97.7 in the overall analysis [24]. DOR from three reviews ranged from 44 to 243, reflecting the heterogeneity of sensitivity and specificity values reported in the primary studies included in the reviews [21,22,28]. In blood specimens, data from two reviews [23,26] with similar prevalence of IA showed consistent results for predictive values: 0.38–0.42 for positive predictive values and 0.96–0.95 for negative predictive values with a single positive test, and 0.67–0.70 for positive predictive values and 0.93–0.94 for negative predictive values for ≥2 positive tests. Higher positive predictive values (0.88 and 0.96) and DOR (135) were found when both PCR and GM were positive [23]. Similar DORs were seen in two reviews (17 and 15 for a single positive specimen, and 30 and 34 for ≥2 positive specimens) [18,23] (Table 4).

The results of the main subgroup analyses to control for sources of heterogeneity are summarized in Table 4. In two reviews of *Aspergillus* PCR performance on BAL specimens [22,28], antifungal prophylaxis significantly reduced sensitivity of PCR; conversely, results of a review in blood specimens did not find substantial differences in sensitivity values, but a significant decrease in specificity values related to anti-mold prophylaxis [27]. As expected, sensitivity was lower in cohort studies compared to case−control studies, and with the degree of adherence to EORTC/MSGERC criteria [21,28].

## 4. Discussion

This umbrella review of SRs was aimed at providing an overall summary of the diagnostic accuracy of PCR-based tests on blood and BAL to diagnose IA in immunocompromised patients. Umbrella reviews collate several SRs on the same topic and consider the inclusion of the highest level of evidence available, such as SRs and meta-analyses [28,29,30]. Umbrella reviews of diagnostic tests provide an opportunity to gain greater insights into test accuracy, as data are summarized across different populations, settings, type of specimen, or other variables, while also considering the overall strength of each study included in the review.

In this umbrella review, the results of eight SRs (twelve records) evaluating the performance of *Aspergillus* PCR tests for the diagnosis of IA in immunocompromised patients, published between 2007 and 2023, were reappraised. The SRs included present data from 167 overlapping reports based on 107 primary studies (98 with a cohort design, 19 with a case−control design) making this, to our knowledge, the largest review of the subject area to date.

When testing BAL specimens, results from five SRs showed a mean sensitivity ranging from 0.57 to 0.91, and mean specificity from 0.92 to 0.97. We graded the level of certainty of these findings as being very low to low due to the risk of bias and due to the heterogeneity (in clinical setting, index test, reference standard adherence, use of antifungal agents) in the primary studies included in the reviews. In blood specimens, results from three SRs showed marked heterogeneity in both sensitivity (ranging from 0.57 to 0.84), and specificity (from 0.58 to 0.95). We graded the level of certainty of these findings as being low (due to the risk of bias and inconsistency) or moderate (due to inconsistency). Lower specificity (0.58) was found in the review of pediatric patients [25], while in two systematic reviews including mostly adult patients, the mean specificity values were higher (0.79 for a single positive test, and 0.93–0.95 for ≥2 positive tests) [23,26]. When pediatric and adult studies are compared, the sensitivity of PCR when testing blood is similar (0.76 in the pediatric review, and from 0.79 and 0.84 for a single positive test, and 0.57 and 0.59 for ≥2 positive tests in the two reviews in adult patients). As with other biomarker tests [31], the use of antifungal therapy was shown to affect performance. In two reviews of BAL specimens [22,28], mold-active antifungals reduced PCR sensitivity, but this effect is not consistent across studies and SRs. Results of a large review in blood specimens [27] did not find substantial differences in PCR sensitivity values, but a significant decrease in specificity values related to mold active agents. It is possible that anti-mold prophylaxis reduces the clinical progression of IA, limiting the manifestations typically associated with IA that are essential when classifying probable IA using the EORTC/MSGERC definitions. Furthermore, given that anti-mold prophylaxis has been associated with reduced GM-EIA sensitivity, the use of mold active agents could result in false-negative GM-EIA results, preventing cases of possible IA being upgraded to probable IA and so compromising the specificity of PCR [27].

There has been considerable progress in standardizing *Aspergillus* PCR protocols and blood-based assays have been shown to be analytically valid. Now it is necessary to consider how the tests can be best used in practice to maximize clinical utility [31]. The use of a standardized PCR may improve performance, and recent evidence suggests that PCR testing in combination with GM-EIA may provide the optimal management strategy [8].

The sensitivity of *Aspergillus* PCR using plasma is superior to that using serum [31,32,33]. PCR positivity occurs earlier when testing plasma and provides sufficient sensitivity for the screening of invasive aspergillosis while maintaining methodological simplicity [32]. However, this level of technical detail has not be assessed in the current SRs. Understanding the influence of *Aspergillus* spp. on PCR performance is also important given analytical sensitivity appears to be reduced when testing for non-*fumigatus* species [34]. Unfortunately, this level of technical detail is generally not provided in the SRs, further compounded by the fact that many cases of IA are diagnosed in the absence of a positive culture or using genus specific tests. While a wide range of both in-house and commercial *Aspergillus* PCR assays are available, performance appears to be comparable [26]. The nucleic acid extraction protocol is critical to optimal performance and is important to understand how variations in this process may be influencing SR data, although currently, the wide variation in combined extraction and amplification protocols may limit/prevent this statistical analysis. Nevertheless, compliance with methodological recommendations when testing blood specimens are associated with improved performance, with significant improvements in specificity [19]. The specificity of *Aspergillus* PCR when testing BAL fluid and when requiring two consecutive positives in blood specimens remains high (>0.92) and subsequent positive likelihood ratios are sufficient to support a diagnosis of IA and the inclusion of *Aspergillus* PCR as a mycological criterion in the 2020 revision of the EORTC/MSGERC definitions for invasive fungal disease [6]. However, clarification on the interpretation of the *Aspergillus* PCR criterion when testing BAL fluid is still needed [35]. While the negative predictive values for *Aspergillus* PCR remain high, the influence of a low pre-test probability (incidence) needs to be observed and the use of negative likelihood ratios better employed (rather than negative predictive values) when test sensitivity is <90%.

As time advances, there will be changes in antifungal prophylactic strategies, treatment of underlying hematological conditions (e.g., CAR-T, monoclonal antibodies), and definitions of invasive fungal disease, which may impact the performance of biomarker assays. These will need to be accounted for when performing future SRs that include both historic and novel datasets. When performing umbrella reviews, the influence of data duplication between individual SR also needs to be considered.

To conclude, this overview summarizes the existing evidence about the diagnostic accuracy of *Aspergillus* PCR assays in immunocompromised patients and allows us to further investigate the evidence available in the existing systematic reviews, assessing variations in study populations, procedures used to conduct the tests, and other variables, with potential to reduce the impact of data heterogeneity by drawing on a broader evidence base. As determined by GRADE assessment, the level of certainty (evidence) for the individual studies included in each SR is variable (very low to low in SRs of PCR on BAL specimens, and from low to moderate in SRs of PCR on blood specimens). However, based on the overall methodological assessment of the SRs included in this umbrella review, in general, we can have high confidence in the methodological quality provided by the SRs, with 75% of SRs meeting or partially meeting all requirements on the AMSTAR-2 checklist and 98.2% of all critical domains being met/partially met across studies.

## Figures and Tables

**Figure 1 jof-09-00967-f001:**
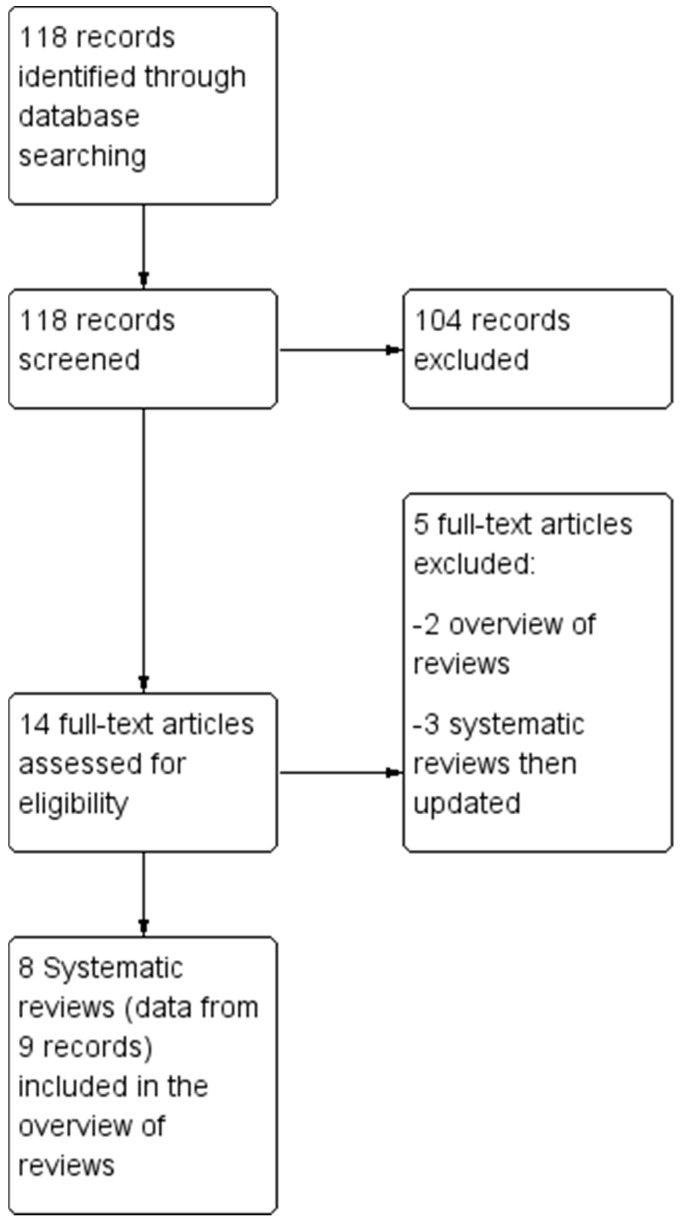
Flow chart of study selection process.

**Table 1 jof-09-00967-t001:** Main characteristics of the systematic reviews (SRs) on *Aspergillus* PCR.

First Author, Year [Ref.]	Clinical Setting	Studies Included in the Review (No. Patients and Specimens)	Diagnostic Test		Quality Assessment	Subgroups Analyses	Main Results
			Index Test	Reference Standard			
Tuon, 2007 [20]	Patients at risk of IA (no further information provided). Control groups included healthy adults or patients without risk factors for IA, patients with high risk for IA, and patients with low risk for IA.	Fifteen trials (seven prospective) from 1995 to 2003; 1232 patients (1308 BAL specimens)	PCR on BAL. The preferred method of PCR was the nested-PCR. Four studies, published after 2001, evaluated qPCR.	EORTC/ MSG 2002	More than 90% of studies met at least 50% of the predefined validity criteria. In eight studies BAL processing was retrospective	Sensitivity was calculated using proven and probable IA. Possible cases not included in the dataset	The overall sensitivity and specificity values of PCR-based techniques in BAL specimens were 79% and 94%
Sun, 2011 [21]	Immunocompromised patients or patients at-risk for IA, mostly haematologic malignancies with pulmonary infiltrates.	Seventeen trials (1991 patients, 1296 BAL specimens) from 1993 to 2009. Nine trials had a case−control design, and eight were cohort studies.	PCR on BAL. BAL collection retrospective in 14 trials.	EORTC/MSG 2002 and 2008	QUADAS-2. The quality of all studies was reported as high (meeting on average 10 of the 14 QUADAS criteria), despite the fact that more than 50% of studies were casecontrol.	Subgroup analyses according to types of PCR, primers (species- vs. genus-specific), study design (cohort vs. case−control), and adherence to EORTC criteria	Six studies used qPCR and the remainder used end-point PCR or semiquantitative PCR.
Avni 2012 [22]	Patients at risk of IA (≥80%)	Nineteen trials (1993–2012), including prospective and retrospective cohort studies and case−control studies; 1585 patients at risk of IA	PCR and GM-EIA in BAL. All studies reported on the diagnostic accuracy of PCR in BAL fluid (10 also on GM in BAL fluid)	EORTC/MSG 2002 and 2008. To avoid incorporation bias, patients in whom the microbiological criterion to define IPA was the GM test were excluded from the analysis	QUADAS-2. Nine studies were at high risk of selection bias. Concerns regarding classification, interpretation and applicability of the reference standard were present in 11 studies	Subgroup analysis according to reference standard definition (EORTC criteria 2002 and 2008), use of anti-mold active agents, type of PCR.	Results were affected by the reference standard and by use of antifungal treatment. No statistically significant differences in the accuracy of qPCR, nested, and other PCRs.
Heng, 2015 [23]	Haematologic patients at risk of IA	Sixteen trials included in the review, but only six trials (402 patients) reported diagnostic data for BAL GM-EIA and *Aspergillus* PCR in individual and combination use. Two studies had a case−control design, and four were cohort studies.	GM alone or with PCR on BAL specimens in six trials. Specimen collection: prospective in two studies, retrospective in four.	EORTC/MSG 2002 and 2008.	QUADAS-2. A significant proportion of studies were at high or unclear risk of bias for different domains of the QUADAS-2 list.	Covariates that may lead to false-positive or false-negative results were not analyzed in this meta-analysis due to paucity of data.	Five studies employed real-time PCR technique and one study used nested PCR. The use of BAL GM-EIA with serum GM-EIA or BAL PCR tests increased the sensitivity moderately when a positive result was defined by either assay. Higher rate of false-positive results to GM-EIA in those receiving beta-lactams at the time of bronchoscopy
Arvanitis, 2015 [24]	Haematologic patients at risk of IA (in 10 trials adult patients, in one trial paediatric patients)	Thirteen trials (three case−control, ten cohort) for a total of 1670 patients. Tests on whole blood and/or serum as weekly screening.	GM and PCR (no information on PCR methods provided). Specimen collection: prospective in seven studies, retrospective in six.	EORTC/MSG 2002 and 2008	QUADAS-2. Most of the studies were of fair quality.	Subgroup analyses according to methodological quality, one or two PCR tests, proven and probable and possible cases vs. proven and probable only, with or without incorporation of GM test.	When screening high-risk patients for IA with GM-EIA and PCR tests, the absence of any positive test has a negative predictive value of 100%, whereas the presence of at least two positive results is highly suggestive of an active infection with a positive predictive value of 88%.
Lerhbecker, 2016 [25]	Invasive fungal disease in pediatric cancer and hematopoietic stem cell transplantation	Twenty-five studies, GM-EIA (*n* = 19), BG (*n* = 3), and PCR (*n* = 11), and 33 comparisons. Retrospective design in eight trials, prospective in seventeen.	GM-EIA, BG, PCR in blood (in one trial BG also in BAL).	EORTC/MSG 2002 and 2008. In four trials, IFI defined by fungal culture, or clinic and imaging, or histology	QUADAS-2. Several studies judged at high risk of bias, particularly selection bias (e.g., case−control studies)	Diagnostic properties are shown both for when possible IFD was included as a negative control and when patients with possible IFD were excluded from the analysis.	All fungal biomarkers demonstrated highly variable sensitivity, specificity, and positive predictive values.
Cruciani, 2019 [26] and 2021 [27]	Immunocompromised patients at risk of IA	Case control excluded; 29 primary studies, corresponding to 34 data sets, published between 2000 and 2018	PCR in blood or serum; 16 studies also evaluated GM assay, but in 15 trials GM-EIA was part of the reference standard. Thus, to avoid incorporation bias, authors did not compare data of GM-EIA assay to PCR (in one trial, sensitivity and specificity were 100% and 96.7% for qPCR, and 88.2% and 95.8% for GM-EIA).	EORTC/MSG 2002 and 2008	QUADAS-2, Most studies were at low risk of bias and low concern regarding applicability.	Subgroup analyses in adult and paediatrics patients, study size, reference standard, requirement of one or more positive specimens to define the test positive, use of anti-mold active agents.	PCR shows moderate diagnostic accuracy when used as screening tests for IA in high-risk patient. Importantly the sensitivity of the test confers a high negative predictive value (NPV) such that a negative test allows the diagnosis to be excluded. AMP significantly decreases *Aspergillus* PCR specificity, without affecting sensitivity.
Han, 2023 [28]	Invasive pulmonary aspergillosis in at risk patients. Most patients had haematologic malignancies Twelve studies (1147 patients.) included primarily patients. With COPD, solid tumor, autoimmune disease with prolonged use of corticosteroids.	A total of 41 studies (5668 patients), including 6 case−control, 20 retrospective cohort and 15 prospective cohorts. Fourteen studies (2061 patients) provided data about proven Invasive Pulmonary Aspergillosis (IPA) only.	PCR in BAL	EORTC/MSG 2002, 2008 and 2020	QUADAS-2. High risk of selection bias in case−control studies	Subgroup analyses showed that the underlying diseases and the use of antifungal treatment had a significant impact on the diagnostic sensitivity of BAL fluid PCR.	BAL fluid PCR is a useful diagnostic tool for IPA in immunocompromised patients and is also effective for diagnosing IPA in patients without HM and HSCT/SOT

**Table 2 jof-09-00967-t002:** Methodological quality of Systematic reviews assessed with the AMSTAR-2 tool.

Author, Year [Reference]		AMSTAR-2 DOMAIN																Overall Confidence in the Results of the SR *
		1	2	3	4	5	6	7	8	9	10	11	12	13	14	15	16	
Tuon, 2007 [20]											na							low
Sun, 2011 [21]											na							high
Avni, 2012 [22]											na							high
Arvanitis, 2015 [23]											na							high
Heng, 2013 [24]											na							high
Lehrnbecher, 2016 [25]											na							low
Cruciani, 2019 [26]											na							high
Han, 2023 [28]											na							high
	Methodological requirement met																	
	Methodological requirement partially met, or not specified																	
	Methodological requirement unmet																	

* We rated overall confidence in the results of the review according to Shea et al. [14], as follows: High, no or one non-critical weakness. Low, one critical flaw with or without non-critical weaknesses.

**Table 3 jof-09-00967-t003:** Summary of findings table (SOT), including the diagnostic accuracy of PCR for *Aspergillus* in BAL and blood specimens. Studies meeting/partially meeting all AMSTAR-2 requirements in bold text.

Author, yr	Specimens, No. Studies	Sensitivity (95% CIs)	Specificity (95% CIs)	GRADE (Levels of Evidence) *
Tuon, 2007 [20]	BAL, 15 studies	0.79 §	0.94 §	Very low (serious RoB, inconsistency)
**Sun, 2011** [21]	BAL, 17 studies	0.91 (0.71/0.96)	0.92 (0.87/0.96)	Very low (serious RoB, inconsistency)
**Avni, 2012** [22]	BAL, 19 studies	0.90 (0.77/0.96)	0.96 (0.93/0.98)	Low (serious RoB)
**Heng, 2015** [24]	BAL, 6 studies	0.57 (0.31/0.80)	0.97 (0.60/1.00)	Very low (serious RoB, imprecision)
**Han, 2023** [28]	BAL, 41 studies	0.75 (0.67/0.81)	0.94 (0.90/0.96)	Low (serious RoB)
**Arvanitis, 2015** [23]	Blood (WB, serum), 13 studies	1 **: 0.84 (0.71/0.92) 2+ ***: 0.57 (0.40/0.72)	0.79 (0.64/0.85) 0.93 (0.87/0.97)	Low (RoB, inconsistency)
Lerherbecker, 2016 [25]	Blood (WB, serum, 1 BAL), 11 studies	0.76 (0.62/0.86)	0.58 (0.42/0.72)	Low (RoB, inconsistency)
**Cruciani, 2019** [26]	Blood (WB, serum), 29 studies	1+ **: 0.79 (0.71/85) 2+ ***: 0.59 (0.40/0.76)	0.79 (0.69/0.89) 0.95 (0.87/0.98)	Moderate (inconsistency)

RoB, risk of bias; CIs, confidence intervals; BAL, bronchoalveolar lavage; WB, whole blood. § CIs not provided. * Very low: The true effect is probably markedly different from the estimated effect. Low: The true effect might be markedly different from the estimated effect. Moderate: The true effect is probably close to the estimated effect. High: The true effect is similar to the estimated effect. ** A single PCR test result was defined as positive. *** 2 consecutive positive specimens are required to define the PCR test positive.

**Table 4 jof-09-00967-t004:** Summary of the measure of diagnostic accuracy other than sensitivity and specificity, and subgroup analyses. Studies meeting/partially meeting all AMSTAR-2 requirements in bold text.

First Author, Year	Other Measures of Diagnostic Accuracy	Subgroup Analyses
Tuon, 2007 [20]	LR+, 10.41 (6.40–16.95); LR−, 0.22 (0.14–0.36)	Different control groups were used in the included studies, but pooled sensitivity and specificity data were only provided for the overall analysis.
**Sun, 2011** [21]	LR+, 11.90 (95% CI, 6.80–20.80); LR−, 0.10 (95% CI, 0.04–0.24). DOR,122 (95% CI, 41–363).	Subgroup analyses showed that the sensitivity was lower with qPCR compared to other types of PCR (mostly nested-PCR and end-point PCR), with species-specific primers compared to genus-specific primers, with cohort design compared to case control, and with degree of adherence to EORTC criteria.
**Avni 2012** [22]	In the overall analysis, NPV, PPV were 97.7/81.6, and DOR 243 (95% CIs, 81–726). In subgroup of cohort studies strictly adherent to reference standard NPV, PPV 94.6/67.7, and DOR 49 (95% CIs, 24–97)	Specificity was uniformly high. Sensitivity was more variable. In nine cohort studies strictly adherent to the 2002 or 2008 EORTC/MSG criteria, sensitivity and specificity were lower compared to overall analysis (77.2%, 95% CIs, 51.5–87.6%, and 93.5%, 95% CIs, 90.6–95.6%, respectively). Antifungal treatment before bronchoscopy significantly reduced sensitivity (58%, 95% CIs, 44.0–70.9).
**Arvanitis, 2015** [23]	1 positive test GM DOR,104 (95% CIs, 37/295) PPV, 61; NPV, 98 PCR DOR, 17 (95% CIs, 7/38) PPV, 38; NPV, 96 GM or PCR, 128 (95% CIs, 37/442) PPV 33; NPV 10 2-positive tests GM DOR, 18 (95% CIs, 7/45) PPV, 59; NPV, 92 PCR DOR, 30 (95% Cis, 13/70) PPV, 67; NPV, 93 GM + PCR, 135 (95% CIs, 38/475) PPV 88; NPV 96	Single positive test results had modest sensitivity and specificity for screening. The screening approach with the highest sensitivity was the one that used at least one GM-EIA or PCR positive result to define a positive episode, achieving a sensitivity of 99%, significantly higher than any single test. Exclusion of low-quality studies from the overall analysis had marginal impact on effect estimates.
**Heng, 2015** [24]	LR pos and neg. provided for BAL GM at different cut-off, but not for PCR, At cut-off of 1, GM LR+,16.1 (95% CIs, 6.2/41.8), LR-, 0.26 (95% CIs, 0.14/0.50), DOR, 61 (95% CIs, 21–181)	Five studies employed real-time PCR technique and one study used nested PCR. The use of BAL GM-EIA with serum GM-EIA or BAL-PCR tests increased the sensitivity moderately when a positive result was defined by either assay. Higher rate of false -positive results to GM-EIA in those receiving beta-lactams at the time of bronchoscopy
Lerhbecker, 2016 [25]	Screening: GM PPV 0–100; NPV, 85–100 PCR PPV, 20–50; NPV, 60–96 Diagnostic: GM PPV 0–100; NPV, 70–100 PCR PPV, 0–71; NPV, 88–100	All fungal biomarkers demonstrated highly variable sensitivity, specificity, and positive predictive values. Poor predictive values for blood GM-EIA, BDG and PCR assays, precluding use as screening tools.
**Cruciani, 2019** [26] **and 2021** [27]	At a mean prevalence of 16%, PPV and NPV were: 1 pos. test 42.8% and 95.1% 2 pos. tests 70.3% and 94.4%. DOR: 1-pos. test, 15.1 (95% CIs, 7.9–28.6 ≥2 tests, 34.5 (95% CIs, 8.2–144.2)	Anti-mold prophylaxis significantly decreased *Aspergillus* PCR specificity (from 86 to 60%), DOR (from 98.06 to 11.80) without affecting sensitivity (83 and 81%). Lower sensitivity and specificity values were found for studies using 2008 criteria compared to those using 2002 criteria: 73.1% (63.2 to 81.1) and 73.3% (60.9 to 82.9) versus 78.7% (70.6 to 85.1) and 82.2% (65.5 to 91.8), respectively (n.s.s.), There was a trend for greater sensitivity and specificity for the in-house assays compared to commercially available kits (0.74 vs. 0.65; 0.84 vs. 0.76, respectively; n.s.s.), Whole blood PCR test had higher sensitivity and lower specificity compared to serum PCR test (n.s.s.).
Han, 2023 [28]	DOR-Neg.LR-Pos.LR 44–11.8–0.27	Sensitivity was lower in prospective, cohort, small group studies and those using revised EORTC/MSG criteria. Antifungal prophylaxis in haematological patients. Reduced sensitivity (from 0.88 to 0.68).

LR, likelihood ratio; DOR, diagnostic odds ratio; PCR, polymerase chain reaction; BAL, bronchoalveolar lavage; NPV, PPV, negative and positive predictive values; CIs, confidence intervals; GM-EIA, galactomannan enzyme immunoassay; n.s.s., not statistically significant.

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
