# Peer review of "An Overview of Systematic Reviews of Polymerase Chain Reaction (PCR) for the Diagnosis of Invasive Aspergillosis in Immunocompromised People: A Report of the Fungal PCR Initiative (FPCRI)—An ISHAM Working Group"

_jof, 2023, doi:10.3390/jof9100967_

Round 1
Reviewer 1 Report
I think it is a well-structured review manuscript that includes the most relevant studies published on this subject. Perhaps the improvement would be to separate cases by species detected, but I understand that this may not be performed in all the studies analyzed in this manuscript.
Author Response
Reviewer #2.
I think it is a well-structured review manuscript that includes the most relevant studies published on this subject. Perhaps the improvement would be to separate cases by species detected, but I understand that this may not be performed in all the studies analyzed in this manuscript.
Au. We thank the reviewer for the positive comment about our overview. With respect to further differentiation to species level, none of the SR’s were able to provide this information. We completely agree that is a relevant point, but as most cases of IA are defined to a probable level using assays that are only genus specific (e.g. GM-EIA) we are unable to perform this analysis. We have included this as a limitation of these studies/this approach
“However, this level of technical detail has not be assessed in the current SRs. Understanding the influence of Aspergillus spp. on PCR performance is also important given analytical sensitivity appears reduced when testing for non-fumigatus species [34]. Unfortunately, this level of technical detail is generally not provided in the SRs, further compounded by the fact that many cases of IA are diagnosed in the absence of a positive culture or using genus specific tests”
Reviewer 2 Report
Invasive aspergillosis (IA) is one of the most common fatal opportunistic fungal infections in immunocompromised patients. Consequently, multiple research studies are constantly being conducted that focus either on developing novel diagnostic tools and or on optimizing pre-existing tests for IA.
The purpose of the review of reviews (termed as umbrella review by the authors) is to reassess the utility of systemic reviews and meta-analyses to evaluate the efficiency of Aspergillus PCR tests for diagnosis of IA. This review is very comprehensive and is easy to follow.
I would like to suggest the following revisions:
1. Please expand the following terms in Table 1: GM-EIA, HM, HSCT/SOT, BAL
2. The authors should briefly explain what are the exclusion criteria used to exclude 104 articles. For example, how many inclusion criteria were missing in these articles before the authors decided to exclude these articles.
English language is fine and only minor editing is required.
Author Response
Comments and Suggestions for Authors
Reviewer #3
Invasive aspergillosis (IA) is one of the most common fatal opportunistic fungal infections in immunocompromised patients. Consequently, multiple research studies are constantly being conducted that focus either on developing novel diagnostic tools and or on optimizing pre-existing tests for IA.
The purpose of the review of reviews (termed as umbrella review by the authors) is to reassess the utility of systemic reviews and meta-analyses to evaluate the efficiency of Aspergillus PCR tests for diagnosis of IA. This review is very comprehensive and is easy to follow.
I would like to suggest the following revisions:
Au. We thank the reviewer for their careful reading of the manuscript and their constructive remarks.
- Please expand the following terms in Table 1: GM-EIA, HM, HSCT/SOT, BAL
Author: Done
- The authors should briefly explain what are the exclusion criteria used to exclude 104 articles. For example, how many inclusion criteria were missing in these articles before the authors decided to exclude these articles.
Author: At the first stage of screening, 14 references met the eligibility criteria, and the full text was examined. 104 records were excluded on first screening of the title and abstract as they did not fulfil our predefined inclusion criteria of being a systematic review on the performance of PCR tests for invasive aspergillosis; the excluded records encompass clinical and diagnostic trials, case reports, and narrative reviews, as now specified in the results section, 2nd paragraph
Comments on the Quality of English Language
English language is fine and only minor editing is required.